# Simultaneous enhancement of strength and conductivity via self-assembled lamellar architecture

Tielong Han [1] ✉, Chao Hou[1], Zhi Zhao [1], Zengbao Jiao [2], Yurong Li[1], Shuang Jiang[3], Hao Lu[1], Haibin Wang[1], Xuemei Liu[1], Zuoren Nie [1] & Xiaoyan Song [1] ✉

Simultaneous improvement of strength and conductivity is urgently demanded but challenging for bimetallic materials. Here we show by creating a self-assembled lamellar (SAL) architecture in W-Cu system, enhancement in strength and electrical conductivity is able to be achieved at the same time. The SAL architecture features alternately stacked Cu layers and W lamellae containing high-density dislocations. This unique layout not only enables predominant stress partitioning in the W phase, but also promotes hetero-deformation induced strengthening. In addition, the SAL architecture possesses strong crack-buffering effect and damage tolerance. Meanwhile, it provides continuous conducting channels for electrons and reduces interface scattering. As a result, a yield strength that doubles the value of the counter-part, an increased electrical conductivity, and a large plasticity were achieved simultaneously in the SAL W-Cu composite. This study proposes a flexible strategy of architecture design and an effective method for manufacturing bimetallic composites with excellent integrated properties.

The pursuit of metallic materials with both high strength and high conductivity has never stopped due to their wide engineering demands[1,2]. Despite a high conductivity, the low strength of pure Cu limits its application in many fields. Thus, people compound Cu with other high-strength phases to form the dispersion strengthened Cu[3] or Cu-X (X can be Ag[4], Ta[5], Nb[6], Ti[7], and W[8]) bimetal composites. In particular, W-Cu composites have attracted considerable attention and play a critical role in numerous key components due to their superior arcing resistance, wear resistance, and high-temperature performance[9,10]. However, the extreme service environment, such as the fields of high-voltage electrical contact, thermonuclear fusion devices, and electromagnetic propulsion system, has put forward higher requirements on both strength and conductivity[11,12]. Most studies to date have mainly focused on strengthening W-Cu composites by solid solution strengthening, secondary phase strengthening[13,14], and grain refinement[15]. Unfortunately, the introduction of solute atoms, phases, and interfaces enhances the electron scattering, resulting in a decrease in conductivity. Therefore, innovative design concepts are urgently needed to develop W-Cu composites with both enhanced strength and conductivity.

Recently, the utilization of heterostructure and bionic structure has been proven to be a viable approach to overcoming the property trade-off paradox[16–21]. For instance, the strength and ductility can be synergistically enhanced by incorporating nanolaminated structures due to the unique constraint effect and microcrack arresting mechanism[22–24]. Moreover, if those laminates are arranged in parallel, a high conductivity may also be achieved, such as in the nanolaminated Cu-Nb composites prepared by accumulative roll bonding (ARB)[25].

[1]College of Materials Science and Engineering, Key Laboratory of Advanced Functional Materials, Ministry of Education of China, Beijing University of Technology, Beijing, China. [2]Department of Mechanical Engineering, The Hong Kong Polytechnic University, Hong Kong, China. [3]Key Laboratory of Electromagnetic Processing of Materials (Ministry of Education), School of Material Science and Engineering, Northeastern University, Shenyang, China. ✉e-mail: tlhan@bjut.edu.cn; xysong@bjut.edu.cn

A synergic enhancement of strength and conductivity is therefore expected if one introduces the lamellar nature into W-Cu composites, rather than only tailoring the composition and grain size. In general, reducing the thickness of the lamella is favorable to enhancing the strength of the laminated composites[4,5,25]. However, the reduction of the lamella thickness would result in a decrease in the electrical and thermal conductivities of the composites. To achieve excellent comprehensive properties, a moderate lamella thickness of several hundreds of nanometers is optimal for the W-Cu composites. Unfortunately, the poor ductility of W makes it difficult to fabricate the laminated W-Cu composites by ARB. The existing laminated W-Cu composites prepared by fast joining of W and Cu foils typically have W laminates with tens of microns thickness[26,27], whose strength is therefore limited due to the absence of interfacial strengthening. While some nanolaminated W-Cu composites prepared by magnetron sputtering exhibit high hardness[28], their conductivity is severely impaired due to the size effect[29]. Moreover, magnetron sputtering is generally not applicable to large volume components necessary for practical applications. Therefore, new strategies for scalable fabrication of lamellar W-Cu composites with suitable lamella thickness are strongly desired. Interestingly, it is found that ultrathin W flakes can be prepared using ball milling[30], which provides a possible approach to the development of laminated W-Cu composites.

In this work, a series of critical challenges are overcome and the laminated W-Cu composite (defined as SAL W-Cu) is architected by a self-assembled scalable powder metallurgy strategy (see details in Methods). As verified by the finite element method (FEM), the SAL architecture is advantageous in enhancing strength and conductivity compared with other types of architectures (Supplementary Note 1). The SAL W-Cu composite features highly aligned W lamellae of submicron thickness with high-density dislocations, which exhibits a

simultaneous enhancement of strength and electrical conductivity (EC), showing advanced integrated properties compared to the counterparts. Unexpectedly, the SAL architecture endows a micro-crack buffering effect and strong damage tolerance to the composite, contributing to a large plasticity. Moreover, the mechanical and electrical responses and underlying mechanisms of the SAL W-Cu are analyzed in depth by combining in-situ high-energy X-ray diffraction (HEXRD), quasi-in-situ compression test and FEM.

## Results

### Construction of SAL W-Cu

There are three key issues to address to construct the architecture of SAL W-Cu. First and foremost, keep the W flakes stacked up parallelly into lamellar architecture. It is noticed that objects with a large diameter-to-thickness ratio (DTR) tend to spontaneously align into parallel architecture under external force. Taking advantage of this, a powder metallurgy-assisted self-assembly approach is proposed, as illustrated in Fig. 1a. In this procedure, spherical W particles with a mean diameter of $14 \pm 3\ \mu m$ are ball milled in ethanol, which produces W flakes with a mean diameter of $55 \pm 7\ \mu m$ and a submicron thickness (Supplementary Fig. 3). The resulting DTR is as large as 100, facilitating the W flakes to automatically orient themselves in the processes of powder loading and pressing.

Second, the W flakes should disperse uniformly into the Cu matrix. Due to the large difference in density between W and Cu and a high DTR of W, it is very difficult to achieve homogeneous mixing while keeping the morphology of flakes using the conventional mechanical methods. The adjacency of W flakes may lead to the occurrence of sintering porosity, which degrades the properties of the sintered composite. By conducting a modified electroless plating method[31], those W flakes are well coated by the nanocrystalline Cu layer, as

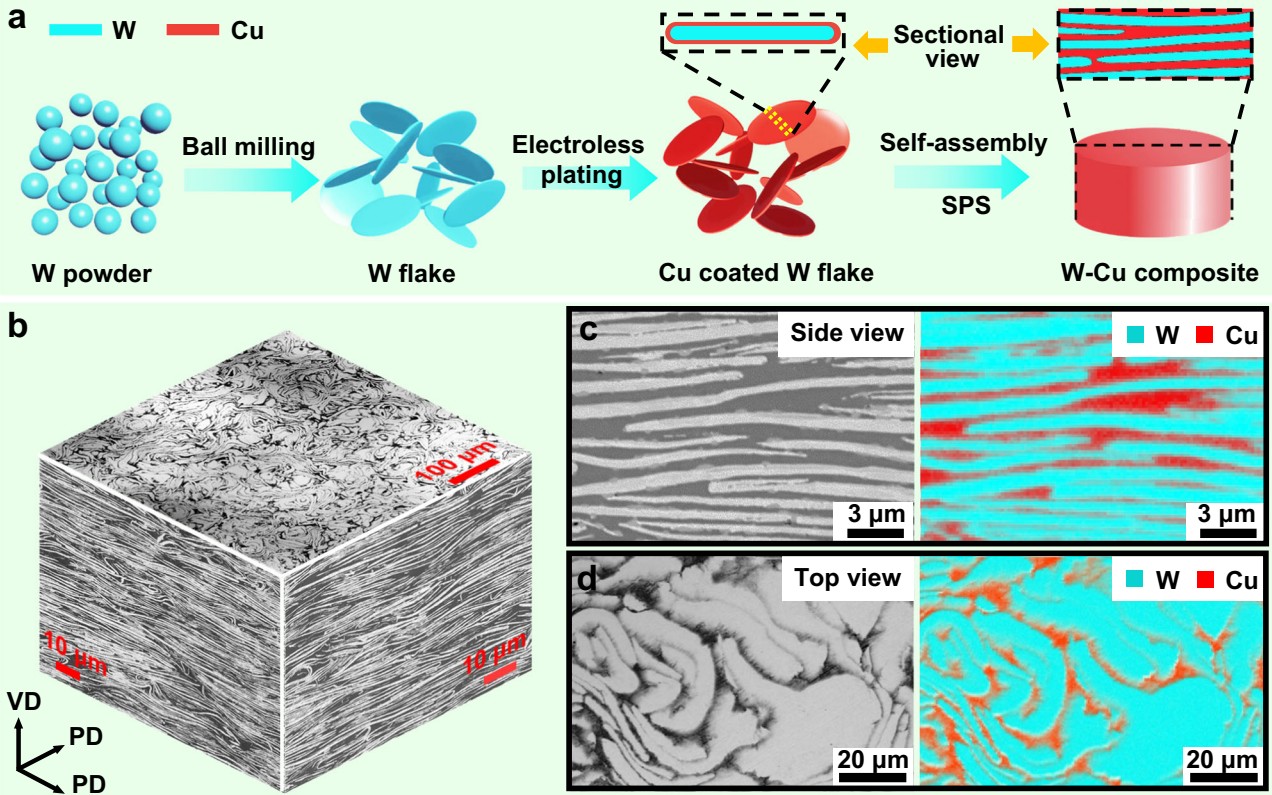

**Fig. 1 | Fabrication strategy and meso-structures of the SAL W-Cu composite.** **a** Schematic diagram of the design strategy for the SAL W-Cu. **b** Three-dimensional reconstructed BSE micrograph of the prepared SAL W-Cu. **c**, **d** Enlarged cross-sectional **c** and top-view **d** BSE images of the SAL W-Cu and the corresponding elemental distribution.

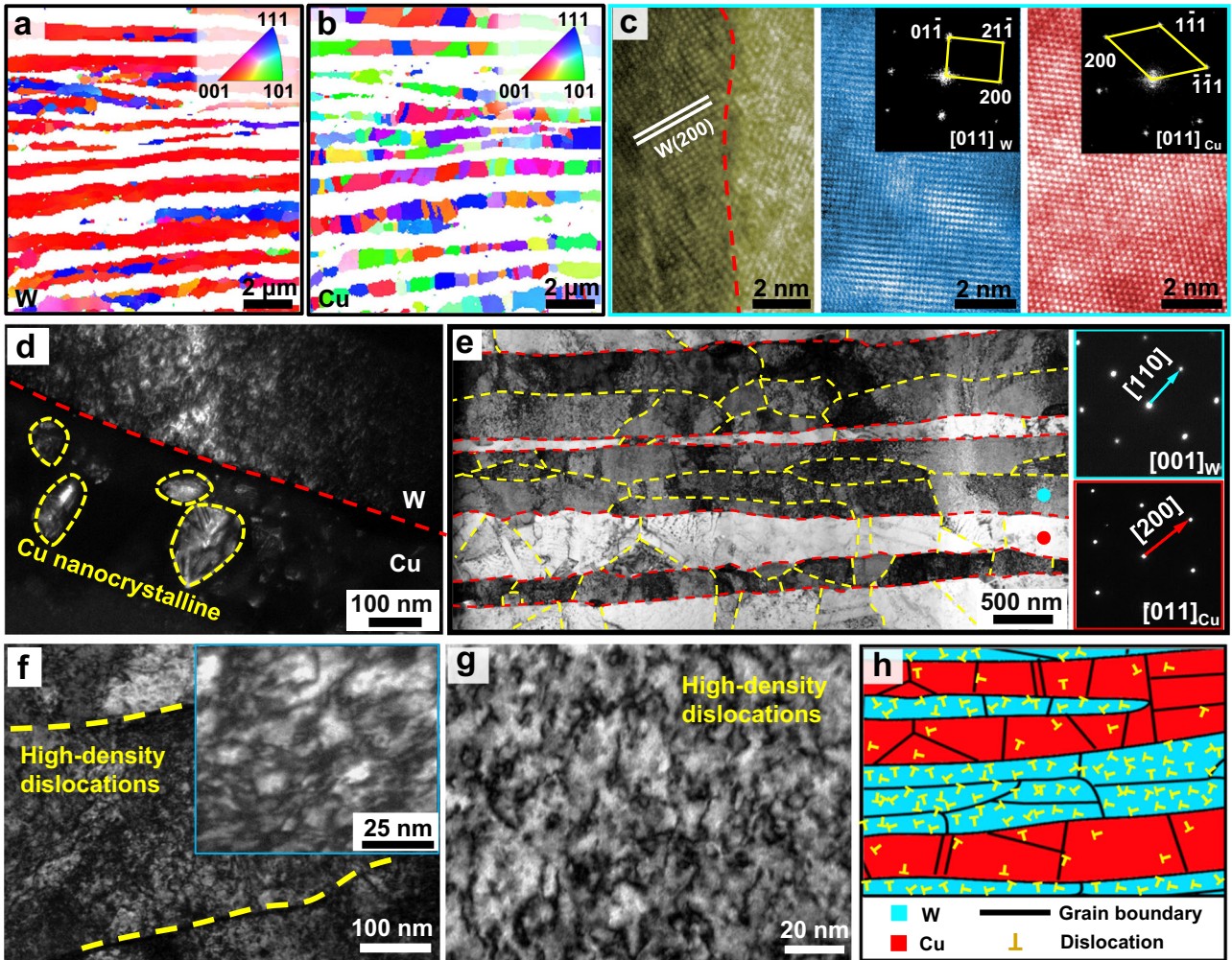

**Fig. 2 | Cross-sectional microstructures of SAL W-Cu composite and electro-lessly plated W-Cu. a** EBSD IPF map of W phase in the as-prepared SAL W-Cu. Inset: map color legend projected on the plane vertical to VD, showing a texture of <100 > w // VD. **b** EBSD IPF map of Cu phase in the as-prepared SAL W-Cu. Inset: map color legend projected on the plane vertical to VD. **c** HRTEM image of the interface in SAL W-Cu. Insets: the corresponding FFT patterns. **d** Dark field TEM image showing the nanocrystalline Cu in the electrolessly plated W-Cu. **e** Bright field TEM image showing the elongated W grains in the SAL W-Cu. Insets: the selected area electron diffraction (SAED) patterns and indexing with the zone axes of [001]W and [011]Cu, respectively. The red and yellow dashed lines represent phase boundaries and grain boundaries. **f** Bright field TEM image of the milled W flake containing dislocations. Inset: an enlarged view. **g** Bright field TEM image of W phase in SAL W-Cu, showing high-density dislocations. **h** Schematic sketch of the SAL architecture in the as-prepared W-Cu composite.

revealed by the backscattered electron (BSE) images and the corre-sponding energy dispersive X-ray spectrometry (EDS) analysis in Supplementary Fig. 3. This effectively promotes the uniform dis-tribution of W and Cu phases in the composite and minimizes the formation of pores in the sintering process.

Last but importantly, to preserve the microstructure of those highly-deformed W flakes, spark plasma sintering (SPS) technique is used for consolidation of the W-Cu composite due to its advantages of rapid heating rate and low sintering temperature. The as-prepared SAL W-Cu composite has a relative density of 97 ± 0.5%, which is at a high level of densification degree among the sintered W-Cu composites[12,31,32]. The SAL W-Cu composite features well aligned W lamellae (Fig. 1b, c). The thickness of each individual W lamella ranges from 100 to 1200 nm, with a mean thickness of 570 nm (Supplementary Fig. 4a). It is worth noting that the W lamellae bend slightly with a statistical average angle of approximately 10° during densification (Supplementary Fig. 4b), and thus the top view reveals a flower-like morphology (Fig. 1d).

To gain a deeper insight into the microstructure of the SAL W-Cu composite, we conducted electron backscatter diffraction (EBSD) and detailed transmission electron microscopy (TEM) characterizations (Fig. 2). EBSD result reveals that the W lamellae in the SAL W-Cu

composite exhibit a <100 > w dominated texture along the vertical direction (VD), i.e. <100 > w // VD, and <110 > w and <100 > w domi-nated texture along the parallel direction (PD), as indicated in Fig. 2a and Supplementary Fig. 5. The same results can also be obtained from the macroscopic X-ray diffraction (XRD) analysis (Supplementary Note 2). This characteristic texture is similar to that generated in the strongly deformed W plates[33], which results from the high-energy milled W flakes. In contrast, Cu exhibits a uniform distribution of ultrafine equiaxed grains with an average size of 718 nm and random orientations (Fig. 2b and Supplementary Fig. 5). Therefore, there is no preferred orientation relationship between W and Cu phases, and most of the phase interfaces are incoherent (Fig. 2c). Compared with the initial nanoscale grain size in the W-Cu powder (Fig. 2d), the grain size of Cu significantly increases. However, further growth of Cu grains is restrained by the ultrafine spacing between the W lamellae. As observed in the bright-field TEM image (Fig. 2e), each W lamella is composed of numerous elongated W grains from the heavy defor-mation during ball milling. The thickness of those W grains varies from 100 to 300 nm while the length is in a range of several microns. Moreover, high-density dislocations are inherited from the original W flakes (Fig. 2f) and retained in the W grains of the SAL W-Cu (Fig. 2g).

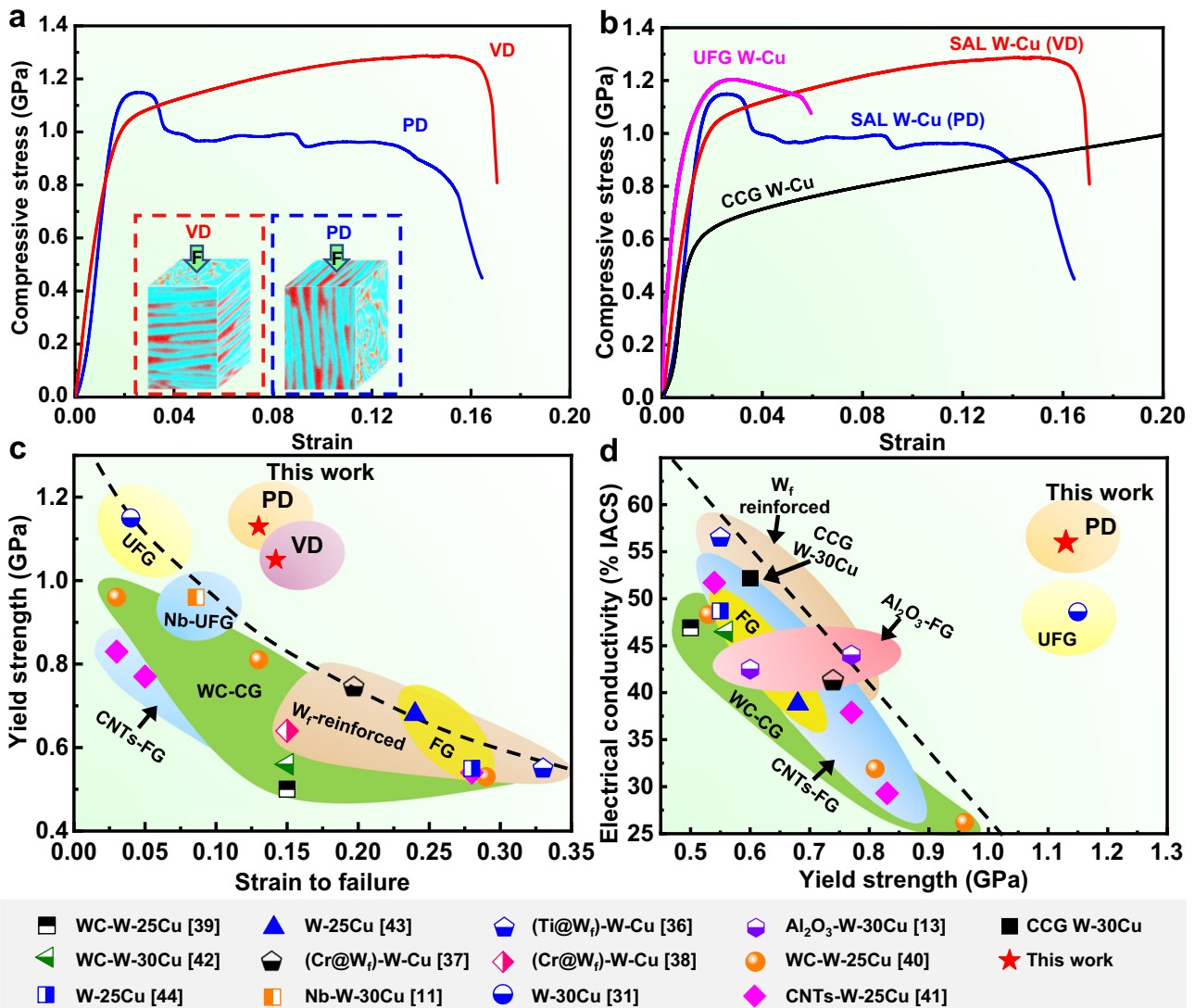

**Fig. 3 | Mechanical properties and electrical conductivity of the as-prepared SAL W-Cu at room temperature. a** Compressive stress-strain curves of SAL W-Cu loaded along VD and PD. **b** Comparison of the compressive curves of SAL W-Cu and other W-Cu composites with a same composition but different grain sizes. **c** Compressive yield strength versus plastic strain of SAL W-Cu and comparison with those of other W-Cu composites. **d** Electrical conductivity versus compressive yield strength of SAL W-Cu and comparison with those of other W-Cu composites. The counterparts include W fiber ($W_f$) reinforced W-Cu[36–38], second-phase enhanced W-Cu[13,39–42], metal-doped W-Cu[11], fine-grained (FG)[43,44] and ultrafine-grained (UFG) W-Cu[31].

## Integrated mechanical and physical properties

Compression tests along both VD and PD were performed to quantitatively evaluate the mechanical performance of the SAL W-Cu, and the typical stress-strain curves are shown in Fig. 3a. The reliability of those mechanical measurements was validated by parallel tests (Supplementary Fig. 8). Compared with commercial coarse-grained (CCG) W-Cu and ultrafine-grained (UFG) W-Cu[31] having the same composition, the SAL W-Cu exhibits better mechanical properties owing to the SAL architecture (Fig. 3b). The yield strength of the SAL W-Cu is

This is because the short sintering duration is insufficient for the recovery process. In addition, due to the difference in thermal expansion coefficients between W and Cu, some dislocations are also present in the Cu grains (Supplementary Fig. 7). As schematically illustrated in Fig. 2h, the presently fabricated SAL W-Cu composite exhibits a peculiar lamellar architecture consisting of alternating fine-grained Cu layers and submicron-thick W lamellae with specific texture and high-density dislocations. W-Cu composites with such structural characteristics have never been reported before.

1130 ± 10 MPa along PD and 1040 ± 15 MPa along VD, which almost doubles that of the CCG W-Cu composite (600 MPa). Although the UFG W-Cu (with a mean grain size of 246 nm) exhibits a similar yield strength as the present SAL W-Cu, it shows a poor plasticity (<6%). By contrast, the SAL W-Cu shows pronounced strain hardening and possesses a larger failure strain of up to 14%, while maintaining an impressive compressive strength of 1300 ± 15 MPa. To the best of the authors' knowledge, the above compressive strength achieves the highest value among the binary W-30Cu composites reported in the literature. Furthermore, when the SAL W-Cu is compressed along PD, the flow stress can still be maintained at a high level (> 0.9 GPa) in a wide range of plastic strains after the first stress drop. This phenomenon has never been reported in conventional double-phase composites. It indicates a high damage tolerance of the SAL W-Cu, which is of great importance for engineering materials to prevent accidents from a sudden rupture in service[34,35].

Comparison of the compressive properties of the SAL W-Cu with other W-Cu composites[11,13,31,36–44] is presented in Fig. 3c. Clearly, the SAL W-Cu exhibits an excellent strength–plasticity combination that

surpasses all the other W-Cu composites. In addition, the present SAL W-Cu also shows a high EC of 56.0 ± 0.3% IACS along PD. This value is higher than that of other W-30Cu composites, and in particular 14% higher than that of UFG W-Cu[31] which has a comparable strength to SAL W-Cu (Fig. 3d). Meanwhile, the SAL W-Cu has a high thermal conductivity of 242 W m$^{-1}$ K$^{-1}$ along PD, and this value is larger than the thermal conductivity of most W-Cu composites with the same composition[45–50] (Supplementary Table 1). Therefore, excellent integrated properties including an ultrahigh strength, high electrical and thermal conductivities, and a good plasticity are simultaneously achieved in a binary W-Cu composite through the SAL architecture construction.

## Discussion

To uncover the origin of the high strength of the SAL W-Cu composite, the progressive yielding behavior of the SAL W-Cu along VD was investigated by in-situ HEXRD (Fig. 4a). Figure 4b shows the interplanar spacing evolution along the loading direction for {200}$_{Cu}$ and {200}$_W$ reflections, which varies in different trends as a function of applied stress ($\sigma$), indicating an evident stress partitioning between W and Cu phases[51,52]. Furthermore, the lattice strain ($\varepsilon_{hkl}$) responses of several typical planes of W and Cu phases are analyzed (Fig. 4c). The deformation process of the SAL W-Cu composite is divided into four stages according to the progressive yielding behavior of Cu and W phases. In the elastic stage ($\sigma < 150$ MPa), all the planes of W and Cu deform elastically, where the lattice strains nearly linearly increase with the applied stress. Subsequently, in the elasto-plastic stage I (with $\sigma$ in a range of 150–600 MPa), the lattice strains of Cu planes gradually deviate from linearity and turn upwards, revealing a progressive yielding of these planes[53]. By contrast, the lattice strain of {200}$_W$ planes remains linear but the corresponding slope changes with respect to the elastic stage, indicating a load transfer from Cu phase to the elastic W phase. In the elasto-plastic stage II (with $\sigma$ in a range of 600–1040 MPa), the lattice strain curve of {200}$_{Cu}$ planes attains yielding, while the lattice strains of {111}$_{Cu}$ and {220}$_{Cu}$ planes further turn upwards. The lattice strain slope of W reflections further deviates from that in elasto-plastic stage I but remains linear, indicating that more stresses are transferred from the Cu phase to W phase. When the applied stress reaches 1040 MPa, the lattice strain curve of {200}$_W$ planes turns upwards and completely yields[54]. Meanwhile, the lattice strains of {211}$_W$ and {110}$_W$ planes deviate and slightly turn downwards, but remain linear due to their higher yield strength, indicating the occurrence of intergranular load transfer in W phase. Since the {200}$_W$ grain family is dominant in W phase along VD, the yielding of {200}$_W$ grain family triggers the yielding of W phase and subsequently leads to the macro-yielding of the SAL W-Cu composite.

Furthermore, the phase-averaged stresses along VD in W and Cu were calculated by the weighted average of the axial stress in different {hkl} oriented grains[55] (Supplementary Note 3). As seen in Fig. 4d, W phase bears significantly higher stress than Cu phase due to uneven stress partitioning, which is consistent with the FEM simulation results (Supplementary Note 1). At the macro-yielding point, W bears a super high stress of 1700 MPa, which is 3.8 times of that of Cu (450 MPa). Thus, the high strength of W lamellae, which stems from the grain refinement and dislocation strengthening, plays a critical role in increasing the yield strength of the SAL W-Cu composite. In addition, it is revealed that the Cu phase undergoes plastic deformation while the W phase remains the elastic state (Fig. 4c), thus the plastic deformation cannot develop sufficiently due to the constraint of the W lamellae[56]. Then the plastic-strain gradients form in Cu phase near the W/Cu interfaces to enable strain continuity, which would further result in the formation of geometrically necessary dislocations (GNDs)[24]. This creates long-range back stress, leading to the resistance against motion of dislocations[18]. Hence, the composite is also strengthened by a hetero-deformation induced (HDI) strengthening. This is further supported by

the results of typical loading-unloading-reloading (LUR) tests (Fig. 4e and Supplementary Fig. 9), where a pronounced hysteresis loop and Bauschinger effect near the yielding point are observed. The HDI stress close to the yielding point is much larger than the effective stress (ES) in the SAL W-Cu composite (Fig. 4e). In contrast, ES occupies a larger proportion in the CCG W-Cu composite (Supplementary Fig. 10). This reveals quantitatively that the HDI strengthening is enhanced in the SAL W-Cu composite due to the increased interface density[24,56]. Therefore, the increased yield strength of SAL W-Cu mainly originates from the exceptional strength of W lamellae and its specific SAL architecture that enhances the HDI strengthening and improves the stress partitioning on the W phase.

Usually, the slope of the stress versus lattice strain (namely the diffraction elastic constant, $E_{hkl}$) varies among different planes due to the elastic anisotropy[53,57]. Thus it can be concluded from Fig. 4c that both the <200>$_{Cu}$ and <200 >$_W$ are the most compliant directions for Cu and W, respectively, while <220>$_{Cu}$, <111>$_{Cu}$ and <211 >$_W$, <110 >$_W$ are the stiffer directions, which is in good agreement with previous studies[58,59]. Since the W lamellae have a strong {200}$_W$ texture (<200 >$_W$ // VD), the proportion of the <200 >$_W$ oriented grains (along the most compliant directions) of the W lamellae along PD is much lower than that along VD. It can be inferred that the yield strength of W lamellae along PD is higher than that along VD, which further results in a higher yield strength of the SAL W-Cu composite along PD.

The high plasticity of the SAL W-Cu along VD is mainly attributed to the sustainable dislocation accumulation and the exclusive crack buffering effect arising from the SAL architecture. As seen in Fig. 3b, the SAL W-Cu exhibits an unchanged strain hardening capability as the CCG W-Cu, indicating a considerable dislocation accumulation during plastic deformation[60]. This is further verified by Fig. 5a$_1$-a$_3$, where a gradually increased density of dislocations and entanglement of dislocations with the increase of strain are observed in Cu. The dislocation density of Cu is increased by 50% during the deformation to failure (Fig. 5b$_1$-b$_2$). In contrast, the dislocation density in W is increased slightly (Fig. 5a$_4$, b$_3$, and b$_4$). This finding indicates the significant contribution of dislocation multiplication in Cu to the high plasticity of the SAL W-Cu. The multiplication of Cu dislocations can be ascribed to the following factors. On one hand, the interface between W and Cu phases can strongly inhibit the dislocation movement[25], thereby facilitating the accumulation of statistically stored dislocations. On the other hand, as indicated by the HEXRD results (Fig. 4c), both Cu and W phases undergo plastic deformation when the stress exceeds the yielding point. However, the Cu phase bears a higher plastic strain than W, leading to the formation of strain gradients, which increase with the plastic strain. Accommodating such strain gradients requires additional GNDs, thereby resulting in a high HDI hardening[23,61], which is evidenced by the gradually increased HDI stress from approximately 518 to 822 MPa with the plastic strain (Fig. 4e). Therefore, both the statistically stored dislocations and the HDI induced GNDs in Cu contribute to the plasticity and the subsequent strain hardening of the SAL W-Cu composite. The HDI induced GNDs play a more important role in the strain hardening than the statistically stored dislocations, as indicated in Fig. 4e.

It is worth noting that plasticity is also closely related to the crack nucleation and propagation stability[22,62]. Generally, stress concentration can easily occur due to the inevitable microstructural inhomogeneity, which usually leads to the nucleation of microcracks. In many materials, once the microcracks initiate, they propagate rapidly and result in a catastrophic fracture of the material in a short duration. This is the main reason that the UFG W-Cu composites show a poor plasticity[31]. In contrast, in the present SAL W-Cu composite, during the quasi-in-situ compression tests, it is interesting to find that a number of microcracks form even at a low strain level (Fig. 5c$_1$-c$_2$), and its density increases gradually with the strain. However, these microcracks do not cause a catastrophic failure of the composite (Supplementary Note 4

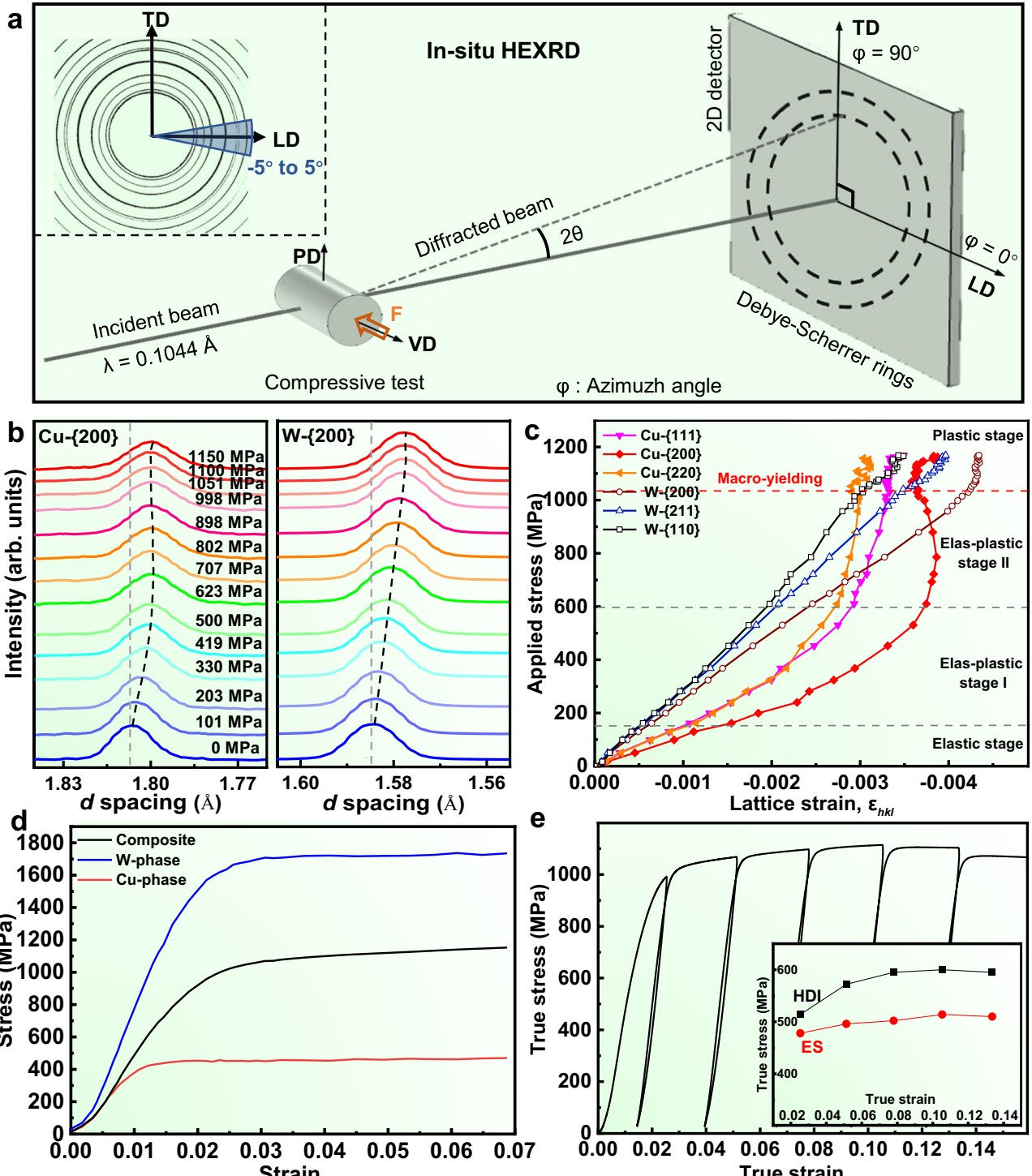

**Fig. 4 | Lattice strains and HDI stress evolution during uniaxial compression along VD. a** Schematic diagram for the setup of in-situ HEXRD compression test. Inset: the azimuth angles in the Debye rings. **b** The corresponding diffraction patterns of {200}$_{Cu}$ and {200}$_W$ at various stress levels along VD according to the in-situ HEXRD results. **c** Evolution of lattice strain versus macroscopic stress for W ({200}$_W$, {211}$_W$, and {110}$_W$) and Cu ({111}$_{Cu}$, {200}$_{Cu}$, and {220}$_{Cu}$) planes along the loading direction. The macroscopic yield strength is marked by the red dashed line. **d** Calculated stress partitioning in W and Cu phases according to the HEXRD results. **e** LUR behavior of the SAL W-Cu along VD. Inset: HDI stress and effective stress (ES) evolution with plastic strain of the SAL W-Cu.

and Supplementary Video). As shown in Fig. 5c₂, the microcracks initiate and propagate mainly along the phase interfaces. It is conjectured that the microcracks are caused by the deformation incompatibility between Cu and W phases. Since the direction of the microcrack propagation (nearly vertical to the loading direction) differs from the direction of the maximum shear stress (with an angle of ~ 45° from the loading direction), the microcracks would not coalesce into a major crack causing fracture. Instead, the formation of such microcracks particularly under high strains helps to release the stress concentrations and strain localization[22,63], which then contributes to the global plastic deformation and thus to the large plasticity of the SAL W-Cu composite. Moreover, as shown in Fig. 5c₃, the angle

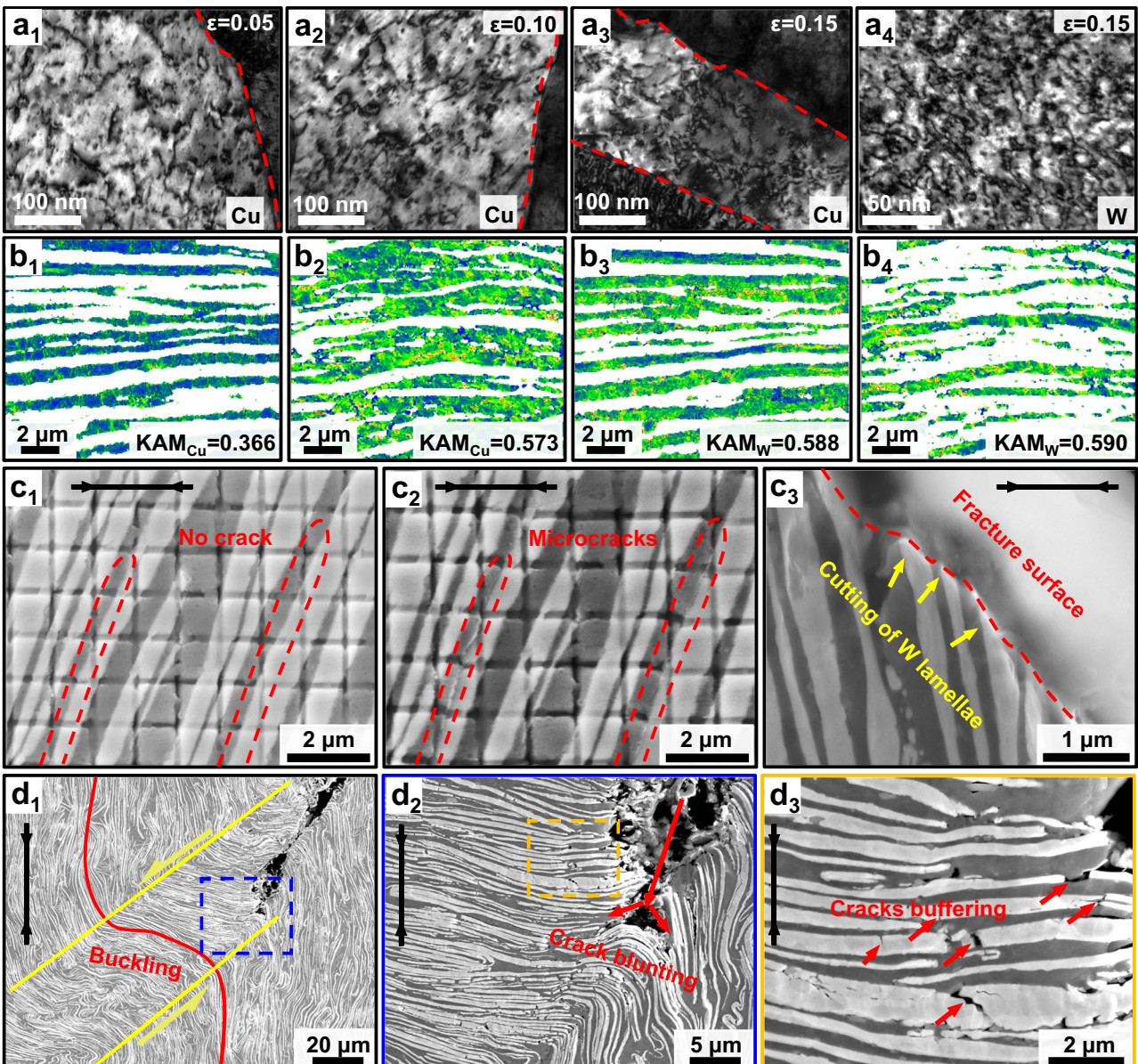

**Fig. 5 | Microstructure evolution during the compression of SAL W-Cu composite.** $a_1$-$a_3$ Bright-field TEM images showing the evolution of dislocation density in Cu phase with the compressive strain (loading along VD). $a_4$ Bright-field TEM image of the W phase with a compressive strain of 0.15 (loading along VD), revealing the high-density dislocations. **b** Kernel average misorientation (KAM) distributions of Cu ($b_1$, $b_2$) and W ($b_3$, $b_4$) phases in the as-prepared composite ($b_1$, $b_3$) and the fractured composite ($b_2$, $b_4$) with loading along VD. $c_1$, $c_2$ SEM images of the composite compressed along VD with a displacement of 0.60 m m and 0.68 mm, showing the appearance of microcracks in the marked regions. $c_3$ Cross-sectional morphology of the fracture surface (loading along VD), showing the cutting of W lamellae. $d_1$-$d_3$ SEM images of the composite compressed along PD, where $d_1$ shows the buckling of W lamellae after the stress drop, $d_2$ is the enlarged view of the selected area in $d_1$, showing the crack blunting, and $d_3$ is the enlarged view of the selected area in $d_2$, showing the cracks buffering in the W lamellae in the kinking band. The black arrows indicate the compression direction.

between the fracture surface and the loading direction is approximately 45°, which implies that the formed main crack has to cut through all the W lamellae in its path of propagation. This means that the SAL architecture and the high-strength W lamellae would increase the critical stress for the formation of the main crack, hence delay the occurrence of fracture and contribute to the large plasticity. Therefore, in addition to the effect of dislocation multiplication on the plasticity of the SAL composite, the crack buffering effect contributes to the plasticity of the composite indirectly at a later stage of deformation.

Furthermore, the deformation mechanisms of the SAL W-Cu along PD were analyzed. As shown in Supplementary Fig. 12, a kinking band with an angle of ~45° towards the loading direction is observed after the first stress drop, and the cracks exist at both ends of the kinking band. The buckling of W lamellae occurs in the kinking band (Fig. 5$d_1$), which is typical for the fiber-reinforced composites under compressive loading due to the elastic instability of the reinforcements[64]. However, as shown in Fig. 5$d_2$, the crack tips are blunted by the lamellar architecture and the cracks are deflected effectively, thereby the propagation of crack is inhibited. The bent W lamellae in the kinking band would be subjected to tensile stress rather than shear stress. As a result, lots of microcracks form in the W lamellae (Fig. 5$d_3$ and Supplementary Fig. 13). However, those emerging microcracks in the W lamellae become arrested at the interface with the alternatingly adjacent Cu phase. The tips of the microcracks are then truncated and the associated high local stresses are shielded[34,65].

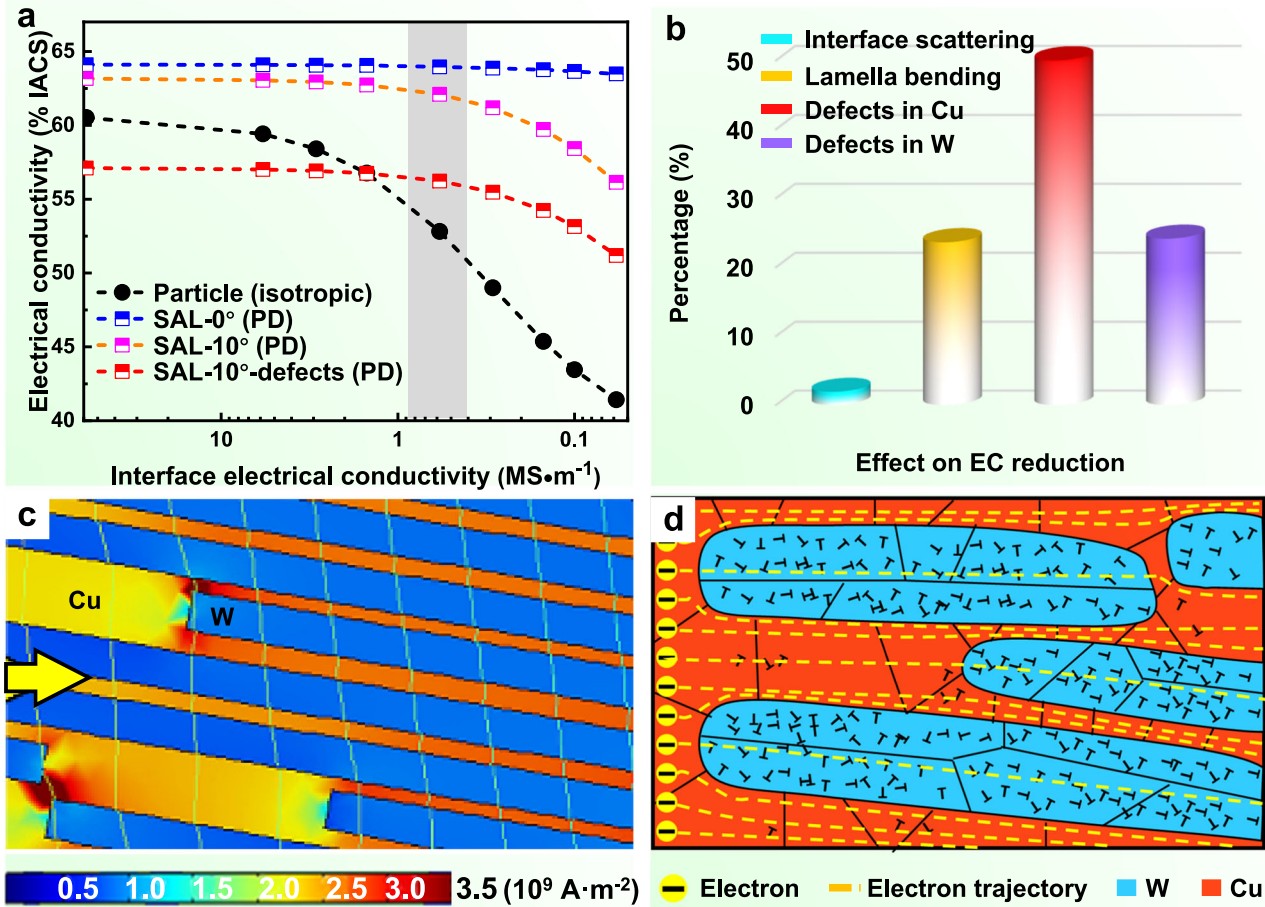

**Fig. 6 | Analysis of the electrical conduction mechanisms of SAL W-Cu composite. a** Evolution of EC as a function of interfacial impedance for different models. "Particle" stands for the W particles reinforced W-Cu, "SAL" stands for SAL W-Cu with the angle representing the degree that W lamella deviates from the horizontal plane, and "defects" indicates voids, dislocations, and grain boundaries. **b** Effects of different factors on the reduction of EC. **c** Distribution of the current density in the SAL W-Cu. The yellow arrow indicates the current direction. **d** Schematic illustration of the electron trajectories in the SAL W-Cu composite.

Therefore, those microcracks remain strictly confined in the individual W lamella where they form, and thus no crack penetrates into the neighboring Cu phase. The formation of microcracks is beneficial to dissipate energy[24,66], thereby releasing the local stress concentrations. The above mechanisms ensure a high stress-bearing capacity of the SAL W-Cu composite along PD. In particular, the SAL W-Cu possesses better crack tolerance compared to the conventional W-Cu composites. It is noted that the stress-strain curve of the SAL W-Cu is fluctuating along PD (Fig. 3a). This is because the composite can be strengthened by the strain hardening of Cu, and new microcracks may form in the W lamellae when the stress reaches a critical value, thus in turn leading to the drop in the flow stress.

The conduction mechanisms of the SAL W-Cu composite were further investigated by FEM simulations (Supplementary Note 5). The influences of W shape, bending angle of W lamellae, defects in W and Cu, and the interfacial resistance on EC of the SAL W-Cu composite were studied by calculations. As shown in Fig. 6a, when the influences of size, interface scattering and defects are not considered, the EC of SAL W-Cu along PD (64.1% IACS, blue dashed line) is 6% higher than that of the W particle reinforced composite (60.5% IACS, black dashed line), indicating that the SAL composite possesses a high intrinsic EC. If the interface scattering effect is introduced, the EC of the particle (with a diameter of 600 nm) reinforced composite decreases seriously due to the highly increased phase interfaces. This is the reason that nanocomposites usually have much lower EC than that of the coarse-grained composites[67]. In contrast, the EC of the SAL W-Cu along PD is

nearly constant even with high interfacial resistance, indicating that the EC of the SAL composite is insensitive to the interfacial resistance. This is mainly because the direction of electron motion is parallel to the interface and thus interface scattering is greatly reduced[25]. It further reveals the advantage of the SAL architecture in improving EC.

The deviation of the experimental EC of the SAL W-Cu from the theoretical value is ascribed to the bending of W lamellae and the defects (such as dislocations, voids, and grain boundaries) in W and Cu phases. As indicated by the pink dashed line in Fig. 6a, the bending of W lamellae reduces EC of the SAL W-Cu along PD, because it increases the interface scattering and aggravates the sensitivity of EC to interfacial impedance. Further, when the grain size, voids, and dislocations are taken into account, the result is plotted by the red line, and it is close to the experimental value of EC at a certain interfacial impedance (the gray area in Fig. 6a). Effects of each influencing factor on the reduction of EC are evaluated in Fig. 6b. As can be seen, the resistance caused by voids and grain boundaries in Cu is the main reason for the decrease of EC along PD, followed by the resistance caused by the defects in W and bending of W lamellae. It is understandable that current prefers to flow in Cu due to its lower resistance, as shown in Fig. 6c. Correspondingly, a schematic diagram of the electron trajectories in the SAL W-Cu composite is shown in Fig. 6d.

The high thermal conductivity of the SAL W-Cu along PD should also be attributed to the unique SAL architecture. The thermal conductivity is mainly composed of electronic thermal conductivity and phonon thermal conductivity, both of which are affected by the defects

such as dislocations, grain boundaries, and phase interfaces [68–70]. As the phonon thermal conductivity is negligible along PD compared to the electronic thermal conductivity in the SAL W-Cu (Supplementary Note 6), the thermal conduction in the composite is essentially dominated by the electrons. Thus, the defects in the composite would also cause the reduction in the thermal conductivity of the SAL W-Cu. Therefore, reducing the bending angle of the W lamellae would further improve the conductivity of the SAL W-Cu composite.

In summary, we proposed a design strategy for high-performance dual-phase materials by engineering a unique SAL architecture, which allows W-30Cu composite to exhibit an ultrahigh yield strength of 1.13 GPa, a high electrical conductivity of 56.0% IACS, and a large plastic strain of 14% at room temperature. The excellent strength stems from the increased stress partitioning to the W phase and the HDI strengthening brought by the SAL architecture, in addition to the strength increase of the W lamellae due to the grain refinement and dislocations strengthening. The sustainable dislocation accumulation capacity induced by the SAL architecture and the crack buffering effect result in a large plasticity of the composite. The high conductivity is also attributed to the specific SAL architecture, which provides a continuous conducting channel for electrons and reduces interface scattering. The SAL architecture strategy and related mechanisms for enhancing the mechanical and physical properties are applicable to the development of metallic composites with excellent integrated properties.

## Methods

### Material preparation

For the preparation of W flakes, spherical W particles with an average diameter of $14 \pm 3\,\mu m$ were firstly mixed with 100 ml ethanol and ball milled in the zirconia jar at 500 rpm for 4 days, with a zirconia-ball to powder ratio of 20:1. To obtain the Cu-plated W flakes, a certain amount of W flakes were added into the plating solution composed of $CuSO_4$ ($21.47\,g\,L^{-1}$), ethylene diamine tetraacetic acid disodium (EDTA-$Na_2$, $80.00\,g\,L^{-1}$), dipyridyl ($0.20\,g\,L^{-1}$), NaOH ($10\,g\,L^{-1}$), HCHO ($53.33\,ml\,L^{-1}$), and ethanol ($333.33\,g\,L^{-1}$). The solution was heated to 333 K and a sodium hydroxide solution was added continuously, maintaining the pH in a range of 11–12. The Cu-plated W flakes were washed for three times with deionized water and then dried. The dried powder was reduced by $H_2$ at 673 K for 1 h. Finally, the reduced powder was loaded in the mold and vibrated, followed by a SPS sintering at 1233 K for 10 min with a pressure of 125 MPa. The heating rate of SPS was $100\,K\,min^{-1}$. The consolidated SAL composite was 20 mm in diameter and 10 mm in height.

### Microstructural characterization

The EBSD analysis and SEM observation were performed with a JSM 7200 F equipped with an EDAX Velocity Super EBSD and EDS detectors. To eliminate the stress and deformation layers caused by mechanical polishing, the samples were ionically polished with Leica EM RES102 prior to SEM or EBSD testing. TEM characterization was conducted on a FEI Tecnai G2 F20 at 200 kV and the samples were prepared by the focused ion beam (FIB). X-ray diffraction (Rigaku Ultima IV, Japan) with radiation of Cu Kα ($\lambda = 1.5406\,Å$) operating at 40 kV and 40 mA was used to detect the crystallographic structure. The thickness and bending angle of W lamellae were measured using ImageJ software.

### Measurements of mechanical and physical properties

The EC was measured by the eddy current conductivity meter (Sigma 2008, Xiamen Tianyan Instruments Co., Ltd). All the samples were mechanically polished prior to testing. The laser flash method was carried out for the measurement of thermal diffusion coefficient ($\alpha$) of the samples on Netzsch LFA 457 MicroFlash along PD at room temperature. The specific heat ($Cp$) of the samples was measured on a

NETZSCH STA 449F3 differential scanning calorimeter (DSC) with a heating rate of $10\,K\,min^{-1}$. The density ($\rho$) was measured according to the Archimedes principle. The thermal conductivity of the composites was calculated according to $K = \alpha\rho Cp$. Room-temperature compression tests were conducted using a LD26 series universal testing machine (LSI, Shanghai) with a strain rate of $3 \times 10^{-4}\,s^{-1}$. Cylindrical samples were machined by electrical discharge machining to a height of 5 mm and a diameter of 3 mm. At least five samples were tested for each condition to obtain statistically reliable results. All compressive tests were performed using a 5-mm extensometer (Epsilon) to monitor strain.

### LUR test

Cylindrical samples with a height of 5 mm and a diameter of 3 mm were used. The strain rate was the same as that used for the uniaxial compression. Upon compression to a designed strain, the sample was unloaded by the stress-control mode to 200 N, followed by reloading to another designed strain before the next unloading.

### Quasi-in-situ compression test in SEM

A dog-bone-shaped compressive sample with a total length of 8.0 mm, a gauge length of 1.5 mm, a width of 2.0 mm, and a thickness of 1.4 mm was used for testing. The loading direction was parallel to the VD of the composite. The strain rate was the same as that in the uniaxial compressive test. Some grids were etched on the surface as markers, and the sample was loaded to a specified nominal strain, then paused and imaged by SEM.

### In-situ HEXRD test

A cylindrical sample with a height of 8.0 mm and a diameter of 3.8 mm was used for the HEXRD test, with the loading direction paralleled to the VD of the composite and at a strain rate of $1.25 \times 10^{-3}\,s^{-1}$. The X-ray beam size was $0.8 \times 0.8\,mm^2$ with an energy of 119 keV (wavelength of 0.1044 Å). The diffraction patterns were collected by a two-dimensional (2D) detector of $2048 \times 2048$ pixels with a spatial resolution of 200 μm (pixel size). A schematic showing the setup of the in-situ HEXRD during compressive test is presented in Fig. 4a. During the test, the beamline was perpendicular to the VD of the composite, i.e. the loading direction. An exposure time of 0.1 s was used to capture the real-time signal during the in-situ test. The distance between the sample and the detector with approximately 1900 mm was calibrated with a $LaB_6$ powder standard. The obtained 2D diffraction patterns were processed using the Fit2D software[54]. The lattice strain was calculated by the formula $\varepsilon_{hkl} = (d_{hkl}-d_0)/d_0$, where $d_0$ and $d_{hkl}$ correspond to the interplanar spacings for different {hkl} reflections without and with external stress, respectively.

## Data availability

All data to evaluate the conclusions are present in the manuscript and the Supplementary Material. The raw data that support the findings of this study are available from the corresponding authors upon request.

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

## Acknowledgements

This work was supported by the National Natural Science Foundation of China (52101031, 92163107, 52371128, 52101032, 52171061, 52101003, 52271085, and 51621003) and the National Key R&D Program of China (2021YFB3501502).

## Author contributions

T.H. and X.S. designed the research; T.H. carried out the main experiments and wrote the main draft of the paper; T.H., C.H., S.J., Y.L, X.L., H.W., and H.L. analyzed the data; Z.J., Z.Z., Z.N., and X.S. revised the paper. All the co-authors discussed the results and commented on the manuscript.

## Competing interests

The authors declare no competing interests.
