## [Peer Review File · Nature Communications]

Simultaneous enhancement of strength and conductivity via self-assembled lamellar architectureREVIEWER COMMENTS

Reviewer #1 (Remarks to the Author):

This is surely an interesting paper which presented a special ultrafine-laminated W-Cu bulk composite with outstanding properties. The new-type W-Cu composites showed obviously higher strength and conductivity at the same time, compared with those W-Cu composites reported in the literature. For the significant enhancement of the properties, the authors studied the impact of the designed laminated structure on the strengthening, toughening, and conduction, as well as the micro-mechanisms. The statements and conclusions were supported by the experimental proofs such as TEM, HEXRD and quasi-in-situ compression tests, and also calculations based on the finite element simulations. Undoubtedly, this study proposed a new strategy for the design of metallic composites possessing simultaneously high mechanical and physical properties, and provided a feasible approach to obtain the special ultrafine-laminated structure. Therefore, it merits publication in Nature Communications, however, clarifications and improvements are required prior to the final acceptance, as listed below.

1. In the Introduction section, the authors presented the mechanical and electrical properties of the W-Cu composites with distinctly different structures produced by the finite element simulations. Surely FEM method can design various structures for the composites and predict part of their mechanical properties. However, my concern is, why only two types of these structures were focused in the present study? While other types, for example, the continuously layered structures, were not considered. In addition, it should be noted that there is obvious deviation between the simulated mechanical properties and the measured values of the prepared composite. The authors are required to give the explanations for the causes of this difference.

2. Page 3, line 73: the authors emphasized that "new strategies are urgently needed for the scalable synthesis of lamellar W-Cu composites with suitable lamella thickness". It is crucial to clarify what is the so-called "suitable lamella thickness", whether it falls within the range of tens or hundreds of nanometers. Is there any approach to design or determine this parameter? How does it affect the structure and properties of the bimetallic composites? As seen in the Fig. S4a, the layer thickness of the as-prepared composite is about 600 nm, how is this value correlated with the "suitable lamella thickness"?

3. A high density is crucial for the applications of W-Cu composites, and it is difficult to fabricate the fully densified W-Cu composite bulk materials due to its intrinsic immiscibility of the bimetal. In this study, the density information was not provided for the specific laminated W-Cu composites. Would the flake-like powder particles cause a decrease in the density of the final composites? Related information should be provided, and the effect of the preparation method of the initial powder on the density of the composite bulk materials should be discussed.

4. When discussing the strengthening mechanisms, the authors declared that the SAL architecture increased the interface density and thus significantly elevated the HDI strengthening effect. Is there any direct evidence to support this statement? In view of this, it would be helpful if the authors carry out the LUR tests for other W-Cu composites with different types of structures and provide the comparison.

5. As for the contribution of dislocations to the plasticity of the composites, the authors identified three types of dislocations in their results. Is it possible to distinguish which is the main contribution of these different dislocations? Or the dislocations generated from different sources have equivalent contributions to the plasticity? In my opinion, the W dislocations may not have a substantial impact. Similarly, when discussing the contributions of dislocations motion and the suppressing of cracks propagation by the laminated structure on the plasticity, the relative importance of these factors should be clarified.

6. The laminated W-Cu composites exhibited impressive thermal and electrical conductivities of the laminated W-Cu composites. However, the analysis was only focused on the mechanisms of the electrical conduction enhancement. It is suggested to add some analysis or discussions of the thermal conductivity mechanisms of this laminated structure.

7. Some minor errors:

Page 1, line 33, the word "of dual-phase composites" should be deleted.

Page 5, line 140, there is a display error in the sentence "i.e. $\{100\}_W \text{ ? VD}$ ", please correct it.

In the Methods section, there is a format error in the temperature unit.

Fig. S13, the quality is not good, better to replace it with a clearer figure.

Reviewer #2 (Remarks to the Author):

The authors succeeded to prepare a W-Cu composite with simultaneous high yield strength (over 1 GPa) high electrical and thermal conductivity and large plastic strain (14%). This is clearly an important step for dual materials, especially since it concerns W and Cu, which are candidate materials for very demanding applications (e.g. fusion reactors, advanced weapon systems, etc.)

Using various high level experimental techniques to investigate the microstructure of the materials they have been able to ascribe the excellent mechanical and transport properties obtained to the unique particularities of this composite. The results are worth publishing in Nature Comm., however I would suggest some corrections before accepting the paper.

1) Firstly, the paper is not very easy to read, since there are key figures in the supplementary material part. I suggest to consider including S7 and S8 in the main paper.

2) On line 144, the authors state "This characteristic texture is inherited from the highly cold-deformed W flakes, which is similar to that found in cold-rolled W plates". One should take into account that the cold rolled plates are initially hot rolled and that it is not clear if the 500 rpm milling does not produce local temperatures above the recrystallization temperature of W (at about 1200 C). Additional comments should be included.

3) As for all severe deformed materials (in this case W), heating the material (prolonged exposure to high T, thermal cycling) is prone to destroy the specific microstructure and therefore to alter the properties. The author should comment on this aspect and provide hints about why they expect this material to be superior in this aspect (if they do so).

Some minor errors:

4) On line 140 there is square symbol, please check and correct.

5) On line 216: "...and a well plasticity" please replace with "good plasticity"

Response to Reviewers' Comments and Questions on Our Manuscript NCOMMS-23-44819

Dear Reviewers,

We appreciate very much for the comments and questions from the reviewers, which are very helpful for improving the quality of our manuscript. We have responded to all the comments and questions from the reviewers point by point. Correspondingly, we carefully revised our manuscript according to the reviewers' comments and suggestions, which were marked in red in the text.

REVIEWER COMMENTS

Reviewer #1 (Remarks to the Author):

This is surely an interesting paper which presented a special ultrafine-laminated W-Cu bulk composite with outstanding properties. The new-type W-Cu composites showed obviously higher strength and conductivity at the same time, compared with those W-Cu composites reported in the literature. For the significant enhancement of the properties, the authors studied the impact of the designed laminated structure on the strengthening, toughening, and conduction, as well as the micro-mechanisms. The statements and conclusions were supported by the experimental proofs such as TEM, HEXRD and quasi-in-situ compression tests, and also calculations based on the finite element simulations. Undoubtedly, this study proposed a new strategy for the design of metallic composites possessing simultaneously high mechanical and physical properties, and provided a feasible approach to obtain the special ultrafine-laminated structure. Therefore, it merits publication in Nature Communications, however, clarifications and improvements are required prior to the final acceptance, as listed below.

We appreciate very much for the reviewer's positive comments.

1. *In the Introduction section, the authors presented the mechanical and electrical properties of the W-Cu composites with distinctly different structures produced by the finite element simulations. Surely FEM method can design various structures for the composites and predict part of their mechanical properties. However, my concern is, why only two types of these structures were focused in the present study? While other types, for example, the continuously layered structures, were not considered. In addition, it should be noted that there is obvious deviation between the simulated mechanical properties and the measured values of the prepared composite. The authors are required to give the explanations for the causes of this difference.*

Thanks for the reviewer's comment. Yes, W-Cu composite models with different structures can be generated by the finite element simulations. However, for the detailed studies in calculations, we only focused on three types of these structures for their mechanical, electrical and thermal properties.

The selection of the three types of structures is based on the representativeness of their structural characteristics and the feasibility of experimental preparation of the corresponding bulk composites and effective refinement of their structure. The finite element simulations proposed possible structures with relatively better mechanical, electrical and thermal properties, but did not include the size effect of the structure on these properties. In experiments, we tend to refine the structures to improve the strength of the composites, thus the feasibility that the structure can be effectively refined becomes more important. For example, for the laminated W-Cu composites fabricated by fast joining of W and Cu foils, the thickness of the W foil can hardly be refined, which strongly restricts the improvement of the composite strength. The case is the same for the W-Cu composites reinforced with W fibers. Therefore, these structures were not used in further simulations in our work.

To make this clearer, we added the related explanations in the Supplementary Information, as shown in Supplementary Note 1 and also presented below.

“Finite element method (FEM) was used to predict the mechanical, electrical and thermal properties of the modeled W-Cu composites with different architectures. According to the distribution characteristics of W phase in the W-Cu composites, generally the architectures can be categorized into five distinct classes. The first class is W-Cu composite reinforced by W particles, where W particles are dispersed in the Cu matrix separately. This type of composite is usually prepared by sintering the Cu-coated W powders. The second class is W-Cu composite reinforced by W skeleton, where W phase forms a connected skeleton, and this type of composite is generally fabricated by melting infiltration. The third class is a hybrid structure, which consists both W particles and W skeleton, and this type of composite is mainly prepared by powders mixing and sintering. The fourth class is W-Cu composite reinforced by W fibers, where W phase is present in the form of fibers. The fifth class is laminated W-Cu composite, where W and Cu layers are distributed alternately.

It is known that refining the structure is beneficial to the strength of the W-Cu composites. Among the five characteristic types of W-Cu composites above, we pay more attention to those whose architectures are feasible to experimentally prepare and refine. Therefore, two of them are selected, i.e., Supplementary Fig. 1a₁ and a₂. Then three representative architectures are evaluated in the following studies:”

Then we would like to give explanations for the difference between the simulated mechanical properties and the measured values of the prepared composite.

As we mentioned in Supplementary Note 1 in the Supplementary Information, the mechanical response of the composites obtained from simulations depends on the input parameters of material properties and structure. The input yield strengths of W and Cu are 900 MPa and 300 MPa, respectively, which are much lower than the actual strengths of W and Cu in

the SAL W-Cu composite (1700 MPa for W and 450 MPa for Cu, as depicted in Fig. 4d in the manuscript). Consequently, the simulated yield strength in Supplementary Note 1 (Supplementary Fig. 2d) is lower than that of the prepared SAL W-Cu composite. However, it is very important to obtain the correlation between the architecture and the properties of the simulated composites by the FEM simulations.

Considering the reviewer's question, we added the related description to the Supplementary Note 1 in the Supplementary Information, which is also presented below.

“As the mechanical response of the composites revealed in the FEM simulations is affected by the input parameters of material properties and structure, the actual strength of the prepared W-Cu composites can be different from the simulated yield strength. However, the inputs are the same for the three models, thus the simulation results are comparable. More importantly, the relationship between the designed architecture and the properties of the composite obtained by the simulations is instructive for subsequent studies.”

2. Page 3, line 73: the authors emphasized that “new strategies are urgently needed for the scalable synthesis of lamellar W-Cu composites with suitable lamella thickness”. It is crucial to clarify what is the so-called "suitable lamella thickness", whether it falls within the range of tens or hundreds of nanometers. Is there any approach to design or determine this parameter? How does it affect the structure and properties of the bimetallic composites? As seen in the Fig. S4a, the layer thickness of the as-prepared composite is about 600 nm, how is this value correlated with the “suitable lamella thickness”?

Thanks for the reviewer's questions. In terms of enhancing the strength of the laminated composites, to reduce the lamella thickness is preferable. In the literature, many studies have shown that the strength of the laminated composites is increased by reducing the lamella thickness (e.g., *Acta Mater.*,

2005, 53: 4817; *Acta Mater.*, 2016, 110: 341; *Mater. Des.*, 2021, 200: 109455 and *Compos. Part B: Eng.*, 2021, 211: 108662). However, reducing lamella thickness would also result in a decrease in the plasticity and conductivity of the composites. For example, the electrical resistance of the laminated Ti-Cu composites increased rapidly when the lamella thickness was below a critical value (*J. Alloy. Compd.*, 2017, 701: 127). The magnetron sputtered W-Cu laminated composite with a lamella thickness of 100 nm had a low electrical conductivity of 34.5% IACS, which was about 30% lower than that of the conventional coarse-grained W-Cu composites (*ACS Appl. Mater. Interfaces*, 2020, 12: 8886). To achieve excellent comprehensive properties, we consider that a moderate lamella thickness, e.g., several hundreds of nanometers, is optimal for the W-Cu composites. It allows to obtain both high strength and conductivity while maintaining acceptable plasticity.

To express this point more clearly, we added the following sentences in the revised manuscript (lines 63-68, page 3).

“In general, reducing the thickness of the lamella is favorable to enhancing the strength of the laminated composites^{4,5,25}. However, the reduction of the lamella thickness would result in a decrease in the electrical and thermal conductivities of the composites. To achieve excellent comprehensive properties, a moderate lamella thickness of several hundreds of nanometers is optimal for the W-Cu composites.”

3. A high density is crucial for the applications of W-Cu composites, and it is difficult to fabricate the fully densified W-Cu composite bulk materials due to its intrinsic immiscibility of the bimetal. In this study, the density information was not provided for the specific laminated W-Cu composites. Would the flake-like powder particles cause a decrease in the density of the final composites? Related information should be provided, and the effect of the preparation method of the initial powder on the density of the composite bulk materials should be discussed.

We agree with the reviewer that it is difficult to obtain a fully densified W-Cu composite bulk material. The densification degree of the W-Cu composites is influenced by the morphology of the sintering powder, the additives, and the sintering process. Generally, high-temperature liquid-phase sintering leads to a higher relative density. However, this processing often results in the formation of coarse grain structures that degrade the mechanical properties of the composite. Addition of the activating elements can improve the densification of the W-Cu composite but may deteriorate the conductivity. In the previous studies, Li et al. prepared the W-Cu composite with a relative density of 96.5% by the liquid-phase sintering (*J. Alloy. Compd.*, 2018, 766: 204). By utilizing the spark plasma sintering (SPS) technique, the W-Cu composite with a relative density of 95.4% and W-Cu-Cr composite with a relative density of 97% were prepared (*Compos. Part B: Eng.*, 2022, 233: 109664; *Int. J. Refract. Met. Hard Mater.*, 2021, 101: 105673). In our work, the as-prepared SAL W-Cu composite had a relative density of $97\pm 0.5\%$, which is at a high level of densification among W-Cu composites reported in the literature.

In addition, it is noted that the use of mixed W and Cu flakes tends to cause a low relatively density of the W-Cu composite (e.g., 91.7% in the previous study in *Materials*, 2022, 15: 7736). The reason is that the plate-like powders are not conducive to the densification process during sintering. Taking account of this point, we designed a special configuration of uniform Cu-coating of W flakes, which facilitated a high densification of the composite after sintering.

According to the reviewer's comment, we added the information of density in the revised manuscript (lines 121-123, page 5 and lines 127-129, page 5), which are also presented below.

“This effectively promotes the uniform distribution of W and Cu phases in the composite and minimizes the formation of pores in the sintering process.”

“The as-prepared SAL W-Cu composite has a relative density of

97±0.5%, which is at a high level of densification degree among the sintered W-Cu composites^{12,31,32}.”

4. When discussing the strengthening mechanisms, the authors declared that the SAL architecture increased the interface density and thus significantly elevated the HDI strengthening effect. Is there any direct evidence to support this statement? In view of this, it would be helpful if the authors carry out the LUR tests for other W-Cu composites with different types of structures and provide the comparison.

Thanks for the reviewer’s suggestion. The LUR test of the commercial coarse-grained (CCG) W-Cu composite has been conducted, with the results presented in the Supplementary Fig. 10. It is observed that the hysteresis loop of the CCG W-Cu composite is less pronounced compared with that of the SAL W-Cu composite, indicating a weakened Bauschinger effect. In addition, the HDI stress proportion of the CCG W-Cu composite is significantly lower than that in the SAL W-Cu composite. These findings strongly support our statement that the SAL architecture increased the interface density and thus significantly enhanced the HDI strengthening effect.

Considering the reviewer’s comment, we added the following content in the revised manuscript (lines 293-297, page 11-12).

“The HDI stress close to the yielding point is much larger than the effective stress (ES) in the SAL W-Cu composite (Fig. 4e). In contrast, ES occupies a larger proportion in the CCG W-Cu composite (Supplementary Fig. 10). This reveals quantitatively that the HDI strengthening is enhanced in the SAL W-Cu composite due to the increased interface density^{24,56}”

Supplementary Fig. 10 a Loading–unloading–reloading curve of the CCG W-Cu composite. **b** Changes of HDI stress (HDI) and effective stress (ES) with the strain of the CCG W-Cu composite.

5. As for the contribution of dislocations to the plasticity of the composites, the authors identified three types of dislocations in their results. Is it possible to distinguish which is the main contribution of these different dislocations? Or the dislocations generated from different sources have equivalent contributions to the plasticity? In my opinion, the W dislocations may not have a substantial impact. Similarly, when discussing the contributions of dislocations motion and the suppressing of cracks propagation by the laminated structure on the plasticity, the relative importance of these factors should be clarified.

Thanks for the reviewer’s questions and suggestions. We agree with the reviewer’s opinion that the limited multiplication of dislocations in W may not play a major role in the plasticity of the composite. Thus, the plasticity and further the strain hardening capability of the SAL W-Cu composite are attributed to the two types of dislocations in Cu, which is evidenced by the greatly increased dislocation density of Cu, as shown in Fig. 5b₁ and b₂. Both the statistically stored dislocations and the HDI induced GNDs in Cu contribute to the plasticity and the subsequent strain hardening of the SAL W-Cu composite. The HDI induced GNDs play a more important role in the strain hardening than the statistically stored dislocations, as indicated in Fig. 4e in the manuscript.

In addition, the crack buffering effect of the SAL architecture contributes to the increase of plasticity indirectly by impeding strain localization hence retaining the plastic deformation of the whole composite. In other words, without the formation of microcracks, fracture failure may occur earlier in the composite. Given the fact that a substantial number of microcracks appear only under high strains, the crack buffering effect manifests at a later stage of plastic deformation.

Considering the reviewer's comment, we added the following content in the revised manuscript (lines 336-338, page 13; lines 349-353, page 14; lines 382-384, page 15).

“This finding indicates the significant contribution of dislocation multiplication in Cu to the high plasticity of the SAL W-Cu. The multiplication of Cu dislocations can be ascribed to the following factors.”

“Therefore, both the statistically stored dislocations and the HDI induced GNDs in Cu contribute to the plasticity and the subsequent strain hardening of the SAL W-Cu composite. The HDI induced GNDs play a more important role in the strain hardening than the statistically stored dislocations, as indicated in Fig. 4e.”

“Therefore, in addition to the effect of dislocation multiplication on the plasticity of the SAL composite, the crack buffering effect contributes to the plasticity of the composite indirectly at a later stage of deformation.”

6. The laminated W-Cu composites exhibited impressive thermal and electrical conductivities of the laminated W-Cu composites. However, the analysis was only focused on the mechanisms of the electrical conduction enhancement. It is suggested to add some analysis or discussions of the thermal conductivity mechanisms of this laminated structure.

Thanks for the reviewer's suggestion. Considering this suggestion, we conducted calculations of the thermal conductivity of W-Cu composites with different architectures by the finite element method, and the new results have been added in Supplementary Fig. 1e and Supplementary Fig. 2d in the

Supplementary Information. The related description concerning the initial and boundary conditions for the thermal conductivity calculations have been provided in Supplementary Note 1 in the Supplementary Information, which are also presented below.

“The thermal conductivities of W and Cu are used as 175 W/m/K and 398 W/m/K⁵, respectively. For the simulations of the thermal conductivity along Z-direction (Supplementary Fig. 1a₁, a₂ and a₃), a temperature of 293.15 K is applied to the surface at Z = 0, and a heat source with a power of 10¹⁰ W/m² is applied to the surface at Z = L_z. For the simulations of the thermal conductivity of the model in Supplementary Fig. 1a₃ along Y-direction, a temperature of 293.15 K is applied to the surface at Y = 0, and a heat source with a power of 10¹⁰ W/m² is applied to the surface at Y = L_y. Thereafter, the thermal conductivities are calculated based on the temperature difference between the two planes (Z = 0 and Z = L_z, Y = 0 and Y = L_y).”

The calculated heat flux distributions in different composite models are shown in Supplementary Fig. 1e, and the predicted thermal conductivities of different architectures are given in Supplementary Fig. 2d. It can be seen that the correlation between the thermal conductivity and the architecture is similar to that between the electrical conductivity and the architecture. The SAL W-Cu composite also has a significantly higher thermal conductivity along the PD direction than other counterparts.

Supplementary Fig. 1 Three-dimensional finite element simulations of the W-30Cu composites with different architectures. **a** Periodic representative volume element models of **(a₁)** particles reinforced architecture, **(a₂)** continuous skeleton architecture and **(a₃)** self-assembled laminated architecture, where the red stands for Cu phase and the cyan stands for W phase. **b-e** The corresponding mises stress distribution in the modeled composites **(b)**, mises stress distribution in W phase **(c)**, current density distribution in the modeled composites **(d)**, and heat flux distribution in the modeled composites **(e)**, respectively. The subscript number (1, 2 and 3) indicates the corresponding architecture and the subscript letter (x, y and z) represents the directions in three dimensions.

Supplementary Fig. 2 Simulation results of mechanical and conductive behaviors. a True stress-strain curves of Cu and W phases used in the simulation. **b** Stress distribution of the W phase with a compressive strain of 0.1. The red indicates the stress level of most of the elements in the model. **c** Compressive stress-strain curves of W-Cu composites with different architectures. **d** Comparison of the compressive yield strength, electrical conductivity and thermal conductivity of different W-Cu composites.

7. Some minor errors:

Page 1, line 33, the word “of dual-phase composites” should be deleted.

Thanks for the reviewer’s correction. The word “of dual-phase composites” has been removed in the revised manuscript.

Page 5, line 140, there is a display error in the sentence “i.e. $\{100\}_w$? VD”, please correct it.

Thanks for the reviewer’s correction. It should be “ \$\langle 100 \rangle_w // VD\$ ”.

In the Methods section, there is a format error in the temperature unit.

Thanks for the reviewer's carefulness. The unit of temperature has been changed to "K" in the revised manuscript.

Fig. S13, the quality is not good, better to replace it with a clearer figure.

Thanks for the reviewer's suggestion. Supplementary Fig. 13 (Supplementary Fig. 12 in the revised Supplementary Information) has been replaced by a clearer figure in the revised Supplementary Information (see below).

Supplementary Fig. 12 Cross-sectional morphology of the SAL W-Cu composite, sampled after the first drop of the stress when loaded along PD. a The global

viewfield. **b-d** Enlargement of local regions in a. Kinking bands and cracks at the ends of the kinking bands are observed.

Reviewer #2 (Remarks to the Author):

The authors succeeded to prepare a W-Cu composite with simultaneous high yield strength (over 1 GPa) high electrical and thermal conductivity and large plastic strain (14%). This is clearly an important step for dual materials, especially since it concerns W and Cu, which are candidate materials for very demanding applications (e.g. fusion reactors, advanced weapon systems, etc.)

Using various high level experimental techniques to investigate the microstructure of the materials they have been able to ascribe the excellent mechanical and transport properties obtained to the unique particularities of this composite. The results are worth publishing in Nature Comm., however I would suggest some corrections before accepting the paper.

We appreciate very much for the reviewer's positive comments.

1. Firstly, the paper is not very easy to read, since there are key figures in the supplementary material part. I suggest to consider including S7 and S8 in the main paper.

Thanks for the reviewer's suggestion. We have moved the Supplementary Fig. 7 and Fig. 8 to the main paper, i.e., we integrated these two figures into Fig. 2 in the revised manuscript, as presented below.

Fig. 2 Cross-sectional microstructures of SAL W-Cu composite and electrolessly plated W-Cu. **a** EBSD IPF map of W phase in the as-prepared SAL W-Cu. Inset: map color legend projected on the plane vertical to VD, showing a texture of $\langle 100 \rangle_{\text{W}} // \text{VD}$. **b** EBSD IPF map of Cu phase in the as-prepared SAL W-Cu. Inset: map color legend projected on the plane vertical to VD. **c** HRTEM image of the interface in SAL W-Cu. Insets: the corresponding FFT patterns. **d** Dark field TEM image showing the nanocrystalline Cu in the electrolessly plated W-Cu. **e** Bright field TEM image showing the elongated W grains in the SAL W-Cu. Insets: the selected area electron diffraction (SAED) patterns and indexing with the zone axes of $[001]_{\text{W}}$ and $[011]_{\text{Cu}}$, respectively. The red and yellow dashed lines represent phase boundaries and grain boundaries. **f** Bright field TEM image of the milled W flake containing dislocations. Inset: an enlarged view. **g** Bright field TEM image of W phase in SAL W-Cu, showing high-density dislocations. **h** Schematic sketch of the SAL architecture in the as-prepared W-Cu composite.

2. On line 144, the authors state "This characteristic texture is inherited from the highly cold-deformed W flakes, which is similar to that found in cold-rolled W plates". One should take into account that the cold rolled plates are initially hot rolled and that it is not clear if the 500 rpm milling does not produce local temperatures above the recrystallization temperature of W (at about 1200 C). Additional comments should be included.

Thanks for the reviewer’s comment. We agree with the reviewer’s viewpoint. Our previous expression was misleading. We meant the characteristic texture in the W flakes was similar with that in the strongly deformed W plates. According to the reviewer’s comment, we have revised this expression in the context as “This characteristic texture is similar to that generated in the strongly deformed W plates³³, which results from the high-energy milled W flakes.” in the revised manuscript (lines 151-153, page 6).

It is correct that “the cold rolled plates are initially hot rolled”. Regardless of the deformation process, in the heavily deformed W plates, a strong texture with $\langle 100 \rangle_w$ parallel to the normal direction (ND) of the rolling plane is often observed (see Fig. R1 below). For comparison, a strong texture with $\langle 100 \rangle_w // \text{VD}$ was found in the W phase of the SAL W-Cu composite, as demonstrated in Supplementary Fig. 5c₁. In addition, a texture with $\langle 111 \rangle_w // \text{ND}$ was also identified in the unidirectionally rolled W, which was weak in our SAL W-Cu composite. This is due to the fact that the high-energy ball milling treatment of W is different from rolling deformation of W, thus the texture formed in the present W flakes is a little different from that in the rolled W sheet.

Fig. R1 Inverse pole figures of the unidirectionally rolled W³².

Supplementary Fig. 5c1 Inverse pole figures of the W phase in the as-prepared SAL W-Cu composite.

For the temperature condition in the milling process, it is necessary to emphasize that the W flakes were prepared by a ball milling process in a liquid medium. As described in the experimental section (line 497), 100 ml ethanol was put into the jar for ball milling, which nearly filled the whole jar. This allowed for an effective control of the temperature during ball milling in the jar. We measured the inner temperature of the jar immediately after the completion of the milling process. It was found that the temperature was only about 60 °C, thus the recrystallization of W grains definitely can not occur in the milled W flakes. Therefore, we mentioned that the W flakes were at the cold deformed state, and this was evidenced by the high-density dislocations existing in the as-prepared W flakes, as shown in Fig. R2 below. Further, with a sintering temperature of 960 °C (much lower than 1200 °C), the recrystallization of W grains does not take place during the sintering process of the W-Cu composite either. As a result, the high density of dislocations was maintained within the W phase in the as-prepared SAL W-Cu composite, as observed in Fig. 2g in the revised manuscript.

Fig. R2 Observation of high-density dislocations in the as-prepared W flakes.

In response to the reviewer's comment, we emphasized that “Spherical W particles with a mean diameter of $14\pm 3\ \mu\text{m}$ were ball milled in ethanol” in the revised manuscript (line 107 page 4).

3. As for all severe deformed materials (in this case W), heating the material (prolonged exposure to high T, thermal cycling) is prone to destroy the specific microstructure and therefore to alter the properties. The author should comment on this aspect and provide hints about why they expect this material to be superior in this aspect (if they do so).

Thanks for the reviewer's comment. We agree with the reviewer that heating the material at high temperatures for a long time or treat the material with thermal cycling is prone to destroy the specific microstructure thus alter the properties of the prepared material. As mentioned in the response to the Point 2 of the reviewer's comments, for the deformed W flakes, after Cu coating with the electroless plating method, a sintering temperature of 960 °C was used to prepare the SAL W-Cu bulk material. During this sintering process, the specific microstructure containing high-density dislocations of the severe deformed W flakes was preserved in the SAL W-Cu composite (Fig. 2g). This supports a high strength of W phase in the prepared bulk composite.

For the as-prepared SAL W-Cu bulk composite, we carried out

experiments on its thermal stability. After being heated up to 900 °C then held for an hour, the microstructure of the material did not reveal obvious change. The hardness of the 900 °C-annealed material was reduced less than 5% compared with that of the as-prepared SAL W-Cu bulk, as shown in Fig. R3 below. Therefore, the SAL W-Cu composite has a high thermal stability till 900 °C. In the literature, Reiser et al. demonstrated that the highly deformed W foils (which had a similar microstructure with that of the deformed W flakes in our work) exhibited a good thermal stability at 900 °C (*Int. J. Refract. Met. Hard Mater.*, 2017, 69: 66), which agreed with our result.

Since the melting point of Cu is about 1080 °C, the service temperatures of W-Cu composites are below this threshold in most of applications. Therefore, we think that the present SAL W-Cu composite possesses good thermal stability and is applicable to a broad range of applications.

Fig. R3 Changes of hardness retention percentage of the SAL W-Cu composite with temperature.

Some minor errors:

4. On line 140 there is square symbol, please check and correct.

Thanks for the reviewer's correction. It should be "<100>_w// VD".

5. *On line 216: "...and a well plasticity" please replace with "good plasticity".*

Thanks for the reviewer's correction. The word has been corrected in the revised manuscript.

Other modifications:

1. According to the journal's formatting instructions, phrases like "novel", "for the first time", "outstanding", and "extremely", have been deleted or replaced by other words in the revised manuscript.

2. According to the journal's formatting instructions, the abstract has been condensed to within 150 words, as presented below.

“Simultaneous improvement of strength and conductivity is urgently demanded but challenging for bimetallic materials. Here we show by creating a self-assembled lamellar (SAL) architecture in W-Cu system, enhancement in strength and electrical conductivity is able to be achieved at the same time. The SAL architecture features alternately stacked Cu layers and W lamellae containing high-density dislocations. This unique layout not only enables predominant stress partitioning in the W phase, but also promotes hetero-deformation induced strengthening. In addition, the SAL architecture possesses strong crack-buffering effect and damage tolerance. Meanwhile, it provides continuous conducting channels for electrons and reduces interface scattering. As a result, a yield strength that doubles the value of the counterpart, an increased electrical conductivity, and a large plasticity were achieved simultaneously in the SAL W-Cu composite. This study proposes a flexible strategy of architecture design and an effective method for manufacturing bimetallic composites with superior integrated properties.”

3. In addition to the revisions according to the reviewers' comments and questions, the language of the whole manuscript was polished carefully.

REVIEWER COMMENTS

Reviewer #2 (Remarks to the Author):

The authors have in my opinion answered most of the points raised by the referee's comments. The revised manuscript is clearly improved. However the new additions have some aspects which should be clarified before publishing the work:

1) main paper, line 463. Although the electronic contribution to thermal conductivity is the dominant term in the case of good conducting metals, in the case of composites with many interfaces it can be expected a larger decrease of thermal conductivity value as compared with the decrease in electrical conductivity. This is more evident for the direction perpendicular to the flake. And the phonon contribution should be even more affected. Moreover, both W and Cu have many dislocations and porosity, respectively. These effects should be responsible also for the big difference among Z and Y directions in the supplementary fig. 2 d). Such effects should be included in the discussion concerning thermal conductivity (requested by the other referee).

2) supplementary file, lines 85 and 88 it is mentioned a heat flux of 10^{10} W/m² for the simulation. This means 10 GW/m², which of course is an unrealistic value. By such values even W will be melted and Cu will be long ago evaporated. The FEM simulation should be checked, and realistic values included.

3) the rebuttal, page 19: I agree with the authors that the available data, including the 1 hour annealing at 900 C point to a good thermal stability. However the recovery temperature for W is around 800 C and longer exposure time (hundreds hours) at 900 C is prone to reduce the number of dislocations stored in W and also affect the W-Cu interfaces. The effects should be checked for particular applications.

4) a small error, in main paper in line 522, for LFA either the name or the number are wrong (457 - MicroFlash, 447 NanoFlash).

With points.

With these clarifications included I consider the paper to be fit for publication in Nat.Comm.

Response to Reviewers' Comments and Questions on Our Manuscript NCOMMS-23-44819A

Dear Reviewer,

We appreciate very much for the comments and questions from you, which are very helpful for further improving the quality of our manuscript. We have responded to all of your comments and questions point by point. At the same time, we revised our manuscript carefully according to these comments and suggestions, which were marked in red in the text.

REVIEWER COMMENTS

Reviewer #2 (Remarks to the Author):

The authors have in my opinion answered most of the points raised by the referee's comments. The revised manuscript is clearly improved. However the new additions have some aspects which should be clarified before publishing the work:

We appreciate very much for the reviewer's positive comments.

1) main paper, line 463. Although the electronic contribution to thermal conductivity is the dominant term in the case of good conducting metals, in the case of composites with many interfaces it can be expected a larger decrease of thermal conductivity value as compared with the decrease in electrical conductivity. This is more evident for the direction perpendicular to the flake. And the phonon contribution should be even more affected. Moreover, both W and Cu have many dislocations and porosity, respectively. These effects should be responsible also for the big difference among Z and Y directions in the supplementary fig. 2 d). Such effects should be included in the discussion concerning thermal conductivity (requested by the other referee).

Thanks for the reviewer's comment. We are sorry for the misleading of supplementary Fig. 2d. In this figure, it looks like that the thermal conductivity decreases more than the electrical conductivity in the Z direction. However, this is caused by the plotting, i.e., the two parameters have different axis ranges in Fig. 2d, where the axis of the thermal conductivity is on the rightmost in pink. To avoid misunderstanding, we modified the coordinate range of the thermal conductivity axis in the revised supplementary Fig. 2d, as presented below.

For details, the electrical conductivity along Z direction (50.7 %IACS) is 79.2% of that along Y direction (64.0 %IACS), while the thermal conductivity along Z direction (246 W/m/K) is 88.2% of that along Y direction (279 W/m/K). That is, the reduction of thermal conductivity is lower than the reduction of electrical conductivity along Z direction.

Supplementary Fig. 2d Comparison of the compressive yield strength, electrical conductivity and thermal conductivity of different W-Cu composites.

In addition, it is worth noting that only the architecture and the conduction direction of W-Cu composites were considered in the simulation (supplementary Fig. 2d). Therefore, the reduction of the electrical and thermal conductivities along Z direction is only ascribed to the architecture factor rather than the influence of dislocations, interfaces and porosity. The study on the effects of interface scattering and defects such as voids, dislocations and grain boundaries in both W and Cu on the

electrical conductivity was provided in supplementary Note 5.

For the relationship between the electronic thermal conductivity (k_e) and electrical conductivity (δ), it is generally described by the Wiedemann-Franz law (N. Stojanovic, et al. Physical Review B, 2010, 82, 075418.), i.e., $k_e/\delta=LT$, where L is the Lorenz factor and T is the absolute temperature. Therefore, the factors influencing the electronic thermal conductivity of the W-Cu composites should be similar to those for the electrical conductivity, which are both related to the motion of electrons.

We agree with the reviewer that the effect of phonon should be considered, since the thermal conductivity (k) of metals is mainly composed of electronic thermal conductivity and lattice thermal conductivity (phonon thermal conductivity, k_{ph}), i.e., $k=k_e+k_{ph}$. For metals, the k_{ph} is generally negligible, for which the accurate calculation has not drawn a common conclusion at present (Z. Tong, et al. Physical Review B, 2019, 100, 144306; N. Stojanovic, et al. Physical Review B, 2010, 82, 075418). In the literature (N. Stojanovic, et al. Physical Review B, 2010, 82, 075418.), it was suggested that the k_{ph} of Cu and W were 22.2 and 42.2 W/m/K at room temperature, which are 5.5% and 24.1% of their total thermal conductivities. Thus, k_{ph} contributes more to k in W than in Cu. As demonstrated by Uher (C. Uher. Thermal conductivity of metals, Springer, Boston, 2004, p.76), though the electronic thermal conductivity remains the dominant contribution to the total thermal conductivity of the metal, the lattice thermal conductivity may occupy a significant fraction of the total thermal conductivity when the metal contains a high density of defects. Because the defect scattering has a much stronger effect on the mean free path of electrons than on that of the phonons, the electronic thermal conductivity decreases more than the lattice thermal conductivity. As a result, the lattice thermal conductivity has a relatively larger proportion of the total thermal conductivity. In other words, although both the k_e and k_{ph} would be affected by the defects, such as dislocations, grain boundaries, and phase interfaces, the effect of defects on k_e is larger than on k_{ph} .

In our work, for the SAL W-Cu composite, as shown in supplementary Fig. 1e_{3y}, the heat flux in Cu is much higher than that in W

when the heat transfers along Y direction, indicating that Cu phase contributes more to the total thermal conductivity. In this case, k_{ph} is negligible, and the total thermal conductivity of the composite is essentially determined by the k_e . The ratio of measured thermal conductivity (242 W/m/K) to electrical conductivity (32.48 MS/m) along Y direction is $k/\delta=7.45$, which is similar to the ratio of the simulated thermal conductivity (279 W/m/K) to electrical conductivity (37.12 MS/m), i.e., $k/\delta=7.51$. In contrast, when the heat transfers along Z direction of the SAL W-Cu composite, as shown in supplementary Fig. 1e_{3z}, the heat flux in both Cu and W are similar. In this case, the contribution of k_{ph} to the total thermal conductivity of the composite is larger, as compared to the condition that heat transfers along Y direction. Therefore, the ratio of measured thermal conductivity (188 W/m/K) to electrical conductivity (19.72 MS/m) along Z direction, $k/\delta=9.53$, is higher than the ratio of the simulated thermal conductivity (246 W/m/K) to electrical conductivity (29.41 MS/m), $k/\delta=8.36$.

Considering the reviewer's comment and suggestion, we added the following discussion concerning the thermal conductivity in the newly revised manuscript (lines 463-472).

“The high thermal conductivity of the SAL W-Cu along PD should also be attributed to the unique SAL architecture. The thermal conductivity is mainly composed of electronic thermal conductivity and phonon thermal conductivity, both of which are affected by the defects such as dislocations, grain boundaries and phase interfaces⁶⁸⁻⁷⁰. As the phonon thermal conductivity is negligible along PD compared to the electronic thermal conductivity in the SAL W-Cu (Supplementary Note 6), the thermal conduction in the composite is essentially dominated by the electrons. Thus, the defects in the composite would also cause the reduction in the thermal conductivity of the SAL W-Cu.”

[68] Tong, Z., Li, S., Ruan, X. & Bao, H. Comprehensive first-principles analysis of phonon thermal conductivity and electron-phonon coupling in different metals. *Phys. Rev. B* **100**, 144306 (2019).

- [69] Uher, C. Thermal Conductivity of Metals. (Springer, Boston, 2004).
- [70] Stojanovic, N., Maithripala, D. H. S., Berg, J. M. & Holtz, M. Thermal conductivity in metallic nanostructures at high temperature: Electrons, phonons, and the Wiedemann-Franz law. *Phys. Rev. B* **82**, 075418 (2010).

Correspondingly, the following content concerning discussion of the thermal conductivity has been added in the revised supplementary information.

Supplementary Note 6

As indicated in Supplementary Note 1, the thermal conductivities of the SAL W-Cu along PD and VD are 279 and 246 W/m/K, respectively, obtained by simulations under the condition that the effect of defects such as dislocations, voids, grain boundaries and phase interfaces on the thermal conductivity was not considered. The measured thermal conductivities along PD and VD are 242 and 188 W/m/K, respectively. The difference between the measured and the simulated values should be ascribed to the scattering effect of defects on the electrons and phonons and the electron-phonon interactions in the composite²². The total thermal conductivity k is mainly composed of the electronic thermal conductivity, k_e , and the phonon thermal conductivity, k_{ph} , i.e., $k = k_e + k_{ph}$ ²³. For metals at room temperature, the k_{ph} is usually much smaller than k_e . It is reported that the k_{ph} of Cu is in a range of 17-22 W/m/K, while the k_{ph} of W is ~ 42 W/m/K^{23,24}. Thus, the contributions of k_{ph} to k are $\sim 5.5\%$ for Cu and $\sim 24.1\%$ for W, respectively. In other words, k_{ph} contributes more to k for W than for Cu. In addition, since both k_e and the electrical conductivity, δ , are closely related to the electrons, their relationship can be described by the Wiedemann-Franz law²⁴: $k_e/\delta=LT$, where T is the absolute temperature, L is the corrected Lorenz factor that is material dependent. Therefore, any factors that influence δ would affect k_e proportionally. Moreover, although defects can also affect k_{ph} , the defect scattering has a much stronger effect on the mean free path of electrons than on that of phonons, k_e decreases more than k_{ph} . As a result, k_{ph} has a relatively larger proportion of k with the effect of defects²⁵. In other words, although both the k_e and k_{ph} would be affected by the defects,

such as dislocations, grain boundaries, and phase interfaces, the effect of defects on k_e is larger than on k_{ph} .

Therefore, if k_{ph} contribution is negligible, one can estimate that the factors influencing δ would affect k proportionally for metals. However, if k_{ph} cannot be neglected, the $k/\delta=(k_e+k_{ph})/\delta$ value would increase. For the conduction along PD in the SAL W-Cu, the measured k/δ (7.45) is similar to the simulated value (7.51), indicating that the contribution of the phonon thermal conductivity is negligible along PD in the SAL W-Cu. In contrast, along VD, the measured k/δ (9.53) is obviously larger than the simulated value (8.36), implying that the contribution of the phonon thermal conductivity should not be neglected along VD.

[22] Dong, L. et al. Thermal conductivity, electrical resistivity, and microstructure of Cu/W multilayered nanofilms. *ACS Appl. Mater. Interfaces* **12**, 8886-8896 (2020).

[23] Tong, Z., Li, S., Ruan, X. & Bao, H. Comprehensive first-principles analysis of phonon thermal conductivity and electron-phonon coupling in different metals. *Phys. Rev. B* **100**, 144306 (2019).

[24] Stojanovic, N., Maithripala, D. H. S., Berg, J. M. & Holtz, M. Thermal conductivity in metallic nanostructures at high temperature: Electrons, phonons, and the Wiedemann-Franz law. *Phys. Rev. B* **82**, 075418 (2010).

[25] Uher, C. Thermal Conductivity of Metals. (Springer, Boston, 2004).

2) supplementary file, lines 85 and 88 it is mentioned a heat flux of 10^{10} W/m² for the simulation. This means 10 GW/m², which of course is an unrealistic value. By such values even W will be melted and Cu will be long ago evaporated. The FEM simulation should be checked, and realistic values included.

We agree with the reviewer's opinion. We carried out new simulations with a power of heat source reduced from 10^{10} to 10^3 W/m². The results showed that the heat flux was reduced in the composites (supplementary Fig. 1e). However, this modification does not change the thermal conductivities in the calculation, nor the heat flux distributions in the W-Cu composites with different architectures. Therefore, the conclusions from the new simulation study are the same as before.

Supplementary Fig. 1e The corresponding heat flux distribution in the modeled composites.

3) *the rebuttal, page 19: I agree with the authors that the available data, including the 1 hour annealing at 900 C point to a good thermal stability. However the recovery temperature for W is around 800 C and longer exposure time (hundreds hours) at 900 C is prone to reduce the number of dislocations stored in W and also affect the W-Cu interfaces. The effects should be checked for particular applications.*

We agree with the reviewer that the recovery effect should be considered for the present SAL W-Cu composite when serving at high temperatures for a long period of time. Therefore, the study on the thermal stability of the SAL W-Cu composite is our ongoing research. The present work is mainly focused on the integrated properties of the SAL W-Cu composite at room temperature, and we believe this new material is promising to more applications when its high-temperature properties are further improved.

4) *a small error, in main paper in line 522, for LFA either the name or the number are wrong (457 -MicroFlash, 447 NanoFlash).*

Thanks for the reviewer's correction. The name should be Netzsch LFA 457 MicroFlash, and it has been corrected in the revised manuscript.

With these clarifications included I consider the paper to be fit for publication in Nat.Comm.

We highly appreciate for all the reviewer's constructive comments and suggestions, and thank you very much for the help.

REVIEWERS' COMMENTS

Reviewer #2 (Remarks to the Author):

I am satisfied with the authors' answers and with the final version of the paper which can now be published.